# Effects of Humic Acids on the Ecotoxicity of Fe_3_O_4_ Nanoparticles and Fe-Ions: Impact of Oxidation and Aging

**DOI:** 10.3390/nano10102011

**Published:** 2020-10-12

**Authors:** Lyubov Bondarenko, Anne Kahru, Vera Terekhova, Gulzhian Dzhardimalieva, Pavel Uchanov, Kamila Kydralieva

**Affiliations:** 1Moscow Aviation Institute, National Research University, 125993 Moscow, Russia; dzhardim@icp.ac.ru (G.D.); KydralievaKA@mai.ru (K.K.); 2Laboratory of Environmental Toxicology, National Institute of Chemical Physics and Biophysics, 12618 Tallinn, Estonia; anne.kahru@kbfi.ee; 3Laboratory of Ecotoxicological Analysis of Soil, Faculty of Soil Science, Lomonosov Moscow State University, 119991 Moscow, Russia; vterekhova@gmail.com; 4The Laboratory of Ecological Functions of Soil, Institute of Ecology and Evolution, RAS, 119071 Moscow, Russia; p.uchanov@gmail.com; 5Laboratory of Metallopolymers, Polymers and Composite Materials Department, Institute of Problems of Chemical Physics RAS, 142432 Chernogolovka, Russia

**Keywords:** magnetite nanoparticles, superparamagnetic iron oxide nanoparticles (SPIONs), ferric ions, ferrous ions, humic acids, aging, bioavailability, protozoa, plants, bioassays, aquatic environment, *Sinapis alba*, *Paramecium caudatum*

## Abstract

The magnetite nanoparticles (MNPs) are increasingly produced and studied for various environmental applications, yet the information on their ecotoxicity is scarce. We evaluated the ecotoxicity of MNPs (~7 nm) before and after the addition of humic acids (HAs). White mustard *Sinapis alba* and unicellular ciliates *Paramecium caudatum* were used as test species. The MNPs were modified by HAs and oxidized/aged under mild and harsh conditions. Bare MNPs proved not toxic to plants (96 h EC_50_ > 3300 mg/L) but the addition of HAs and mild oxidation increased their inhibitory effect, especially after harsh oxidation (96 h EC_50_ = 330 mg/L). Nevertheless, all these formulations could be ranked as ‘not harmful’ to *S. alba* (i.e., 96 h EC_50_ > 100 mg/L). The same tendency was observed for ciliates, but the respective EC_50_ values ranged from ‘harmful’ (24 h EC_50_ = 10–100 mg/L) to ‘very toxic’ (24 h EC_50_ < 1 mg/L). The ecotoxicity of Fe-ions with and without the addition of HAs was evaluated in parallel: Fe (II) and Fe (III) ions were toxic to *S. alba* (96 h EC_50_ = 35 and 60 mg/L, respectively) and even more toxic to ciliates (24 h EC_50_ = 1 and 3 mg/L, respectively). Addition of the HAs to Fe-ions yielded the respective complexes not harmful to plants (96h EC_50_ > 100 mg/L) but toxic to ciliates (24 h EC_50_ = 10–100 mg/L). These findings will be helpful for the understanding of the environmental fate and toxicity of iron-based NPs.

## 1. Introduction

Magnetite (Fe_3_O_4_) is the most magnetic, naturally existing mineral. Along with the development of nanotechnologies, nanosized magnetite has increasingly been produced and studied for applications in various fields from medicine [1] to environmental remediation [2]. For example, Fe_3_O_4_ nanoparticles (NPs), due to their small size, high surface-area-to-volume ratio, the possibility for surface modification and excellent magnetic proper, ties have potential in wastewater treatment as magnetically removable low-cost sorbent carriers for phosphorus [3], heavy metals [4] or photocatalytic particles [5,6]. However, the widespread use of iron oxide nanomaterials will inevitably lead to increased environmental emissions. Thus, the fate and environmental effects of iron oxide nanomaterials must be evaluated [7,8]. Often, commercial NPs are surface-coated to increase their colloidal stability. The latter, however, makes them more motile in an aqueous environment [9]. Moreover, upon release to the environment, the NPs will be transformed being exposed to dissolved oxygen, dissolved organic matter (DOM), various ions, etc. [10]. Thus, the initial physicochemical properties of the magnetite NPs will be altered, depending on environmental compartment they will be released to and assumingly there will be a continuous change in the structure and composition of iron oxide NPs in time, ultimately affecting their bioavailability and toxicity. Thus, bare magnetite NPs are susceptible to air oxidation [11], leading to transformation to other forms of iron oxides [12] and become easily aggregated in aquatic media [13]. Schwaminger et al. [12] simulated the oxidation process of magnetite NPs in harsh (0.07 mol/L HNO_3_) and mild (60 °C, air atmosphere) oxidation conditions and showed that in 24 h not only harsh conditions, but also mild conditions led to complete phase transformation of magnetite (Fe_3_O_4_) to maghemite (γ-Fe_2_O_3)_. In particular, under ambient conditions Fe^2+^ on the surface of Fe_3_O_4_ NPs tended to rapidly oxidize leading to the transformation of magnetite into maghemite.

The sensitivity to oxygen often limits the use of magnetite NPs, since the particle size, magnetic properties [8], and toxicity will change [10]. Thus, the transformation processes would alter the chemical composition and structure of the NPs and influence their aggregation, dissolution, transport behavior, and even their potential toxicity [14,15]. Lei et al. [10] showed that the inhibition of growth of algae *Chlorella pyrenoidosa* by iron-based NPs decreased with oxidation of the NPs following the order of nZVI > Fe_3_O_4_ NPs > Fe_2_O_3_ NPs, whereas the toxicity also depended on the crystal phase as α-Fe_2_O_3_ NPs were more toxic to algae (96 h EC_50_ = 71 mg/L) than γ-Fe_2_O_3_ NPs (96 h EC_50_ = 132 mg/L). With similar particle size (20–30 nm), iron-based NPs with higher oxidation induced lower oxidative stress and thus lower toxicity to algae [10].

An important role in the fate and transport of iron oxides in the environment plays the humic substances (HS). HS make up the natural organic matter in soils and sediments and are formed from dead organic matter by microbial degradation. HS is divided into three fractions: humic acids, fulvic acids, and humin and can undergo a number of reactions with Fe, e.g., form complexes with both Fe(II) and Fe(III) via carboxyl groups of the organic matter [6]. Also, HS can (i) sorb to iron oxide particles [4] changing their surface charge and affecting their aggregation and bioavailability [16]; (ii) interfere with mineral dissolution/precipitation reactions [17]; and (iii) drive redox reactions [18]. The adsorption of HS to the surface of engineered NPs may strongly influence, and in some cases control, their surface properties and aggregation behavior.

Recent research has shown that humic acids (HA) have a high affinity to Fe_3_O_4_ NPs and the sorption of HA on the Fe_3_O_4_ NP enhanced the stability of Fe_3_O_4_ nanodispersions by preventing their aggregation via electrostatic and steric effects [19,20,21,22,23]. However, weak Coulombic attraction, hydrogen, and hydrophobic (van der Waals, π-π, CH-π) bonds between Fe_3_O_4_ and HA [24] seemed not to protect Fe_3_O_4_ NPs from the oxidation in real environmental conditions [25,26]. Thus, redox-active HS play an important role in Fe redox speciation in magnetite and can influence their reactivity and role in biogeochemical Fe cycling. Furthermore, such processes may also influence the fate of other elements bound to the surface of Fe minerals. Redox reactions between HS and the mixed-valent mineral magnetite can potentially lead to changes in Fe (II)/Fe (III) stoichiometry and even dissolve the magnetite [25].

Analyzing the Mössbauer spectroscopy data, Bogart et al. [27] proposed the following mechanism of transformation of Fe_3_O_4_ to γ-Fe_2_O_3_ due to the release of Fe-ions from the particles: Fe^2+^ are drawn from the nucleus by a concentration gradient arising due to surface oxidation; Fe^2+^ is oxidized in situ, colliding with mobile electrons, then Fe^3+^ ions are distributed to maintain the electroneutrality of the material. Dissolution of the solid phase and an increase in dissolved Fe^2+^ and/or Fe^3+^ as a result of incubation of NPs with humic substances was demonstrated by Sundman et al. [25]: the magnetite incubated with native humic substances became more oxidized as compared with the control, i.e., bare magnetite.

Dissolution (shedding of metal ions) as a toxicity mechanism of metal-containing NPs has been extensively studied, especially for some types of NPs, such as nanosilver, ZnO NPs and CuO NPs [28]. For example, Navarro et al. [29] evaluated the toxicity of Ag^+^ and AgNPs towards algae *Chlamydomonas reinhardtii* and showed that toxicity of AgNPs to algae was due to shed Ag-ions and particles contributed as a ‘carrier’ and a source of the Ag-ions. Heinlaan et al. [30] showed that the toxic effects of ZnO and CuO NPs to crustaceans *Daphnia magna* and *Thamnocephalus platyurus* and bacteria *Vibrio fischeri* were due to solubilized Zn^2+^ and Cu^2+^. The same conclusion was reached for algae *Pseudokirchneriella subcapitata* exposed to ZnO and CuO NPs by Aruoja et al. [31]. Also, Franklin et al. [32] observed that Zn^2+^ ions released from the ZnO NPs were highly toxic to algae *P. subcapitata*. 

A meta-analysis of the scientific literature made by Bondarenko et al. [33] covered the (eco)toxicological data for Ag, CuO, and ZnO NPs and respective soluble salts on for algae, crustaceans, fish, bacteria, yeast, nematodes, protozoa and mammalian cell lines. The analysis showed that as a rule, crustaceans, algae and fish proved most sensitive to the studied NPs and at least for AgNPs and ZnO NPs the toxicity was fully explained by solubilized Ag- and Zn-ions, respectively. Analogously, Notter et al. [34] made a meta-analysis of ecotoxicity data for the same types of NPs and the respective metal ions showing that, as a rule, NPs proved less toxic than respective dissolved metal ions. However, to our best knowledge, the information on ecotoxicity of Fe_3_O_4_ NPs [35] and especially on the contribution of shed Fe-ions to the overall toxicity and effect of humic substances as toxicity modulators [36] is still limited. The current paper aims to fill that gap.

## 2. Materials and Methods

### 2.1. Synthesis and Modification of Fe_3_O_4_ Nanoparticles

#### 2.1.1. Synthesis of Fe_3_O_4_ and Fe_3_O_4_/HA Nanoparticles

The bare and humic acid (HA)-coated magnetite nanoparticles (Fe_3_O_4_ and Fe_3_O_4_/HA NPs) were synthesized with methods described in [37]. Briefly, a commercial sodium salt of humic acids (HA) (Powhumus, the total acidity of the HA was 5.3 mmol/g of acidic COOH and OH-groups, weight-average molecular weight Mw was 9.9 kD; Humintech, Grevenbroich, Germany) was used for the preparation of Fe_3_O_4_/HA and other chemicals were from Sigma-Aldrich (Sigma-Aldrich Chemie GmbH, Steinheim, Germany). For the synthesis of NPs, 6.1 g of FeCl_3_·6H_2_O and 4.2 g of FeCl_2_ 4H_2_O was dissolved in 100 mL water and heated to 40 °C, then two solutions, 10 mL of ammonium hydroxide (25%), and 0.7 g of HAs were added rapidly and sequentially. The mixture was stirred at 1000 rpm at 40 °C for 10 min under argon atmosphere and then cooled to room temperature. The black precipitate of Fe_3_O_4_/HA NPs was collected by Nd-magnet (0.3 T) and washed to neutral with distilled water (90 °C) and dried under vacuum at 40 °C. The bare Fe_3_O_4_ magnetic nanoparticles were synthesized in a similar way, except that no HA was added.

#### 2.1.2. Simulation of Oxidation of Fe_3_O_4_/HA NPs in Mild and Harsh Conditions

Mild oxidation conditions (aging) were simulated by incubation of the stock aqueous suspensions of Fe_3_O_4_/HA in the dark at 5 °C for 90 days. Harsh oxidation conditions were simulated by mechanical dispersion of the Fe_3_O_4_/HA samples in planetary ball mill where the dispersion process takes place between the grinding balls sliding on each other and between the vessel sides and the grinding beads. The Fe_3_O_4_/HA powder was placed in a wolfram carbide cell with wolfram carbide balls (ball-to-sample mass ratios was 7:1) and dispersed in a high-energy ball mill (SPEX SamplePrep 8000 Mixer/Mill, Metuchen, The Netherlands) at 1425 rpm for 10 min.

#### 2.1.3. Preparation of Complexes of HA with Fe^2+^ and/or Fe^3+^

For the preparation of Fe-HA complexes, a stock solution of the HA (50 mg/L) was prepared by dissolving powdered HA in deionized water with drop-wise addition of 0.1 M NaOH till a final pH = 8. Then, Fe(II) and/or Fe(III) chloride stock solutions (FeCl_2_·4H_2_O, FeCl_3_∙6H_2_O, 1000 mg/L) were added to HA stock (pH 8.0), to obtain the final ratio HA:Fe(II)/Fe(III) in each series of 1:0.15 according to [38].

Altogether nine different preparations were used for biotesting: aqueous suspensions of Fe_3_O_4_, Fe_3_O_4_/HA, HA, soluble salts of Fe^3+^, Fe^2+^, Fe^3+^/Fe^2+^, complexes Fe(II)HA, Fe(III)HA, Fe(II, III)HA. The pH of the suspensions to be tested was adjusted to 7 by diluted HCl and NaOH.

### 2.2. Characterisation of the Microstructure of Magnetic NPs (MNPs)

The phase composition and primary particle size of the Fe_3_O_4,_ Fe_3_O_4_/HA and Fe_3_O_4_/HAox (after oxidation) were determined by X-ray diffraction analysis (XRD) in the Bragg–Brentano geometry using a Philips X-pert diffractometer (Philips Analytical, Eindhoven, The Netherlands, Cr-Kα radiation, λ = 2.29106 Å). The full width at a half maximum (FWHM) of all reflections was used for particle size determination with the Scherrer equation. In order to quantify the oxidation progress, the (440) reflection was fitted with five different functions in Origin 2019 Pro (OriginLab Corporation, Northampton, MA, USA).

The lattice parameters determined for all samples formulated in this study are smaller than those reported for magnetite 8.396–8.400 Å (ICDD–PDF 19–629), but larger than those for maghemite 8.33–8.34 Å (ICDD–PDF 39–1346). A plausible explanation of this phenomenon could be the process of partial oxidation of Fe^2+^ during drying and/or storage and modification resulting in the non-stoichiometric Fe_3-δ_O_4_ formation where δ can range from zero (stoichiometric magnetite) to 1/3 (completely oxidized) [39]. For magnetite with an ideal Fe^2+^ content (assuming the Fe_3_O_4_ formula), the mineral phase is known as stoichiometric magnetite (x = 0.50). As magnetite becomes oxidized, the Fe^2+^/Fe^3+^ ratio (formula 4) decreases (x < 0.50), with this form denoted as nonstoichiometric or partially oxidized magnetite [39]. The stoichiometry can easily be converted to the following relationship (1):(1)$$x=\frac{{\mathrm{Fe}}^{2+}}{{\mathrm{Fe}}^{3+}}=\frac{1+3\delta }{2+2\delta }$$

### 2.3. Characterisation of the Magnetic Properties of Magnetite NPs (MNPs)

Magnetic properties of MNPs dry powders were characterised with Vibrating Sample Magnetometer Lake Shore (Lake Shore Cryotronics, Westerville, OH, USA) at 300 K.

### 2.4. Analysis of the Surface Charge and Hydrodynamic Diameter of Magnetic NPs (MNPs)

Dynamic light scattering (DLS) measurements were conducted with a “Zetasizer 2c” and “Autosizer 2c” equipment (Malvern Panalytical Ltd., Malvern, UK) at 633 nm with a solid-state He–Ne laser at a scattering angle of 173° at 25 °C. For the DLS analysis, each sample was diluted to approximately 0.1 g/L. Prior to the analysis the magnetic NP suspensions were ultrasonicated for 10 s followed by 100 s of standstill. The average values of the hydrodynamic diameter of NPs were calculated from third-order cumulant fits of the correlation functions using Correlator K7032-09 (Malvern Panalytical Ltd., Malvern, UK). The range of pH was ~7. The experiments were performed at constant ionic strengths 0.01 M set by NaCl.

### 2.5. Ecotoxicity Testing of Magnetic NPs (MNPs) and Fe-Ions

The toxicity of aqueous suspensions of magnetic NPs towards ciliates was analysed in the concentration range from 0.33 to 33 mg/L and towards plants *Sinapis alba* from 16.5 to 3300 mg/L. The exact concentrations/dilutions tested depending on the test and MNPs and are indicated in the Figures and Tables. The toxicity values (EC_50_) for MNPs and humic acids (HA) are presented as mg compound/L (nominal concentrations). Prior testing NP suspensions prepared in distilled water were dispersed by ultrasonication using the sonication bath (100 W, 40 kHz) (Heb Biotechnology, Shaanxi, China) for 10 min.

The exposure concentrations of the Fe^2+^, Fe^3+^, Fe^2+^/Fe^3+^ and Fe(II)HA, Fe(III)HA, Fe(II,III)HA used for ecotoxicity testing ranged from 0.149 mg Fe/L to 770 mg Fe/L for ciliates and from 1.49 mg Fe/L to 1540 mg Fe/L for plants. The exact concentrations/dilutions tested depended on the test and compounds. The toxicity values (EC_50_) for Fe-ions are presented as mg Fe/L.

#### 2.5.1. *Paramecium caudatum* Acute Toxicity Test

The ecotoxicity of magnetic NPs (MNPs) and Fe-ions to ciliates were determined using *Paramecium caudatum Ehrenberg* acute toxicity test performed following the protocol described in [40]. Briefly, the assay is based on the measurement of the mortality of *Paramecium caudatum* when exposed to toxic substances compared with the control. The assay was performed in the 96-well polystyrene plates; well’s size 1 mL; Eppendorf). Stock cultures of *P. caudatum* were maintained in the mineral Lozin-Lozinskiy nutrient medium of the following composition, mg/L: NaCl—100.0, KCl—10.0, CaCl_2_∙2H_2_O—10.0, MgCl_2_∙6H_2_O—10.0, NaHCO_3_—20.0 (Sigma-Aldrich Chemie GmbH, Steinheim, Germany). The stock cultures were maintained at room temperature (22 ± 2 °C), pH 7.5–8.0, without any organic compounds added. To start the test culture, about 1/3 of the stock culture was transferred into a Petri dish containing fresh nutrient medium and incubated at 22 ± 2 °C in the dark for 24 h.

Using a stereoscopic microscope (Model MC-1, Micromed, Shanghai, China), 10–15 ciliates were transferred using a capillary pipette into each of 3–4 test wells containing fresh incubation medium. The volume of liquid when transferring ciliates into the wells did not exceed 0.02 mL. In general, each series of wells (control and test wells) contained at least 30 ciliates. 0.6 mL of incubation medium was added to the control wells; 0.6 mL of the test sample was added to the test wells. The plates with samples and ciliates were incubated at 22 ± 2 °C in the dark. During the exposure period, no food or any other supplements were added. After 24 h of incubation, the individuals were checked for the viability in each well using a stereoscopic microscope. Freely moving ciliates were considered viable, and immobilized individuals were considered dead. The mean values were calculated and compared with the control values.

#### 2.5.2. *Sinapis alba* L. Acute Toxicity Test

The toxicity of MNPs and Fe-ions to plants was measured using the white mustard *Sinapis alba* L. root growth inhibition assay (ISO 18763:2016 [41]) in Phytotoxkit format [42]. Certified, high-quality and commercially available seeds were used for all experiments. Following the Phytotoxkit test format (plate test), the 10 mL of pre-shaken NPs suspension of Fe-ions solutions were poured onto transparent test plates (21 × 15.5 × 0.8 cm covered by a white filter paper, and ten *Sinapis alba* seeds were placed on the paper. The test plates were closed with a transparent lid and incubated first in a horizontal position at 20 ± 2 °C in darkness for 24 h, and then for 72 h in a vertical position at 24 ± 2 °C and an illumination period of 16 h per day with a light intensity of 4000–7000 lx (light wavelength 400–700 nm, universal white). At the end of the incubation, the length of the main root of the mustard seedlings was measured. The mean values were calculated and compared with the control values. The test was made in three replicates.

### 2.6. Statistical Analysis

The inhibitory effect of tested compounds/dilutions compared to the control was calculated as a percentage (%). From dose-response curves, the EC_50_ values (mg/L or mg Fe/L, depending on the compound) were calculated using the probit method (IBM SPSS Statistica 17.0, IBM, Armonk, NY, USA) and expressed as an average value ± standard deviation (SD). ANOVA was used for the analysis of statistically significant variances within and between the test groups. The degree of statistical significance of the results was calculated in the R-studio application. The programs were created in the programming language R (inter-group statistical significance was fixed at *p* ≤ 0.05). 

## 3. Results and Discussion

### 3.1. Microstructure of Magnetite Nanoparticles (MNPs)

The crystalline structure of the synthesized MNPs was identified using XRD analysis (Figure 1).

The XRD patterns were similar for all studied NP samples and can be interpreted as a face-centered cubic (fcc) lattice with the parameters of 8.383(2), 8.382(6), 8.365 (5) and 8.250(8) Å for the Fe_3_O_4,_ Fe_3_O_4_/HA, Fe_3_O_4_/HA harsh oxidation and Fe_3_O_4_/HA mild oxidation samples, respectively (Table 1). Finally, the composition of the crystalline component of the pieces can be assigned as follows: Fe_2.94_O_4_, Fe_2.93_O_4_ and Fe_2.84_O_4_ for the Fe_3_O_4,_ Fe_3_O_4_/HA and Fe_3_O_4_/HA harsh oxidation samples, respectively. There was no Fe_3_O_4_ in the Fe_3_O_4_/HA sample after mild oxidation in distilled water during 90 days of aging. Magnetite was likely completely oxidized to maghemite and/or other iron species (Fe(OH)_3_, FeOOH, etc.). According to [43], the XRD patterns confirm preservation of the spinel structure during oxidation processes: the content of magnetite decreased from ~82.7% to ~79.2%, ~48.3% and 0% for Fe_3_O_4,_ Fe_3_O_4_/HA and Fe_3_O_4_/HA harsh oxidation and Fe_3_O_4_/HA mild oxidation, respectively (Figure 2).

A slight decrease in the magnetite content in the samples containing humic acids (Fe_3_O_4_/HA) from ~82.7 % to ~79.2 % (Table 1) can be associated with phenol and quinoid units in the HA structure [44]. Notably, the Fe_3_O_4_ content in the Fe_3_O_4_/HA harsh oxidation sample was almost halved, assumingly due to the oxidation of magnetite NPs during mechanical treatment (harsh oxidation). The latter shows that HAs were not forming a strong protective shell to the core interacting mainly via Coulombic attraction and hydrogen bonds. Thus, the magnetite gradually oxidized, producing the increasing maghemite shell after each stage of treatment, i.e., first during the treatment with HA and then during the oxidation in harsh conditions in the high-energy ball mill. Magnetite nanoparticles are seemingly very sensitive to oxygen, and in the presence of air, may undergo oxidation to Fe(OH)_3_ [45], or to maghemite (γ-Fe_2_O_3_) phase. Small amounts of O_2_ in water could easily oxidize the Fe^2+^ species to Fe^3+,^ becoming a favorable environment for the production of Fe(OH)_3_ or γ-Fe_2_O_3_. Depending on pH of the aqueous solution containing Fe^3+^ ions, goethite (α-FeOOH) may be formed due to hydrolysis [46]. Full oxidation of magnetite to maghemite was observed in the case of 90 days of aging in mild condition in the presence of dissolved oxygen. This is coherent with the Tombacz et al. [47], where solid phase transformation of magnetite to maghemite and formation of akageneite shell on the magnetic core after storage at 4 °C for 6 in aqueous medium was observed by XRD analysis.

The step-wise oxidation of magnetite is schematically depicted in Figure 2.

The coherent-scattering region size was derived from powder XRD data by Scherrer’s method. The full width at half maximum (FWHM) of the reflections was used for particle size determination. In order to quantify oxidation, progress the (440) reflection was fitted with Pseudo-Voight function for Fe_3_O_4_ and Fe_3_O_4_/HA and Voight function for Fe_3_O_4_/HA_OX_ in Origin 2019 Pro. While the spherical particle shape remained constant for all modification routes, a slight particle growth during modification with HA can be observed (Table 1). The diameter of the bare magnetite particles according to XRD analysis was 6.9 nm, i.e., in the size range of superparamagnetic iron oxide NPs (SPIONs) with a high saturation magnetization and a high specific surface area [48,49]. The diameter for Fe_3_O_4_/HA particles was 10.3 nm and that of the Fe_3_O_4_/HA harsh and mild oxidation particles 7.8 and 11.03 nm, respectively. All MNPs were polydisperse [50]. According to the coefficient of variation CV and standard deviation value σ of samples, Fe_3_O_4_/HA had a smaller size distribution (12.6%, 1.3) than Fe_3_O_4_ (34%, 2.4)_,_ Fe_3_O_4_/HA harsh (24.6%, 1.9) and mild (50.2%, 5.1) oxidations (Table 1). Therefore, considering the values of CV and σ it could be concluded that in the oxidation process, the primary size for the NPs practically remained unchanged.

### 3.2. Evaluation of the Magnetic Properties of the Studied MNPs

The most crucial property of magnetite NPs allowing a variety of applications is their ferrimagnetism. Some magnetic characteristics for the Fe_3_O_4_ MNPs are presented in Table 2. The hysteresis loops are closed and symmetrical versus origin of the coordinate system (Appendix A). The shape of the loops evidenced the ferromagnetic character of the material desirable for their application in separation. The respective saturation magnetizations of bare Fe_3_O_4_ and Fe_3_O_4_/HA were 68.2 and 30.9 emu/g, respectively, suggesting the content of HA in Fe_3_O_4_/HA about 40% (w/w). The saturation magnetization for samples of MNPs indicates that magnetite nanoparticles stabilized with humic acids exhibited superparamagnetic properties at room temperature (Appendix A). The reduced saturation magnetization for Fe_3_O_4_/HA to 30.9 emu g^−1^ compared to bare magnetite can be explained by a disordered spin canted structure near the surface of NPs. Our harsh oxidation experiment revealed decreasing to 15.7 emu/g in saturation magnetization as well as changes in the shape of the magnetisation curve (Appendix A). This behavior can be attributed to a change of the composition and to the structure defects arising during oxidation, as shown above by XRD studies (Figure 1). In the absence of a magnetic field, all samples showed a similar low residual magnetism around ±4–7 emu g^−^^1^ due to magnetic viscosity for superparamagnetic materials [51]. The further increase of the coercivity can be related to an increasing anisotropy by phase transformation to maghemite [52].

The black aqueous suspensions of Fe_3_O_4_/HA nanoparticles were oxidized to brown suspensions after storage in distilled water for 90 days, indicating the HA coating was not able to protect the magnetite from oxidation and to maintain its saturation magnetization. Due to that, the magnetic properties of aged Fe_3_O_4_/HA were not measured.

### 3.3. The Ecotoxicity of Bare Fe_3_O_4_ NPs (MNPs) and Humic Acids-Modified MNPs Before and After Oxidation in Mild and Harsh Conditions

The bare MNPs evaluated for the current study’s toxic effects were not toxic to plants *S. alba* in the root length inhibition test: the EC_50_ value was not reached even at 3300 mg/L (Table 3). Literature data support the not harmful nature of Fe-oxide NPs. For example, Fe_2_O_3_ NPs were not inhibitory in the seed germination test as at 1000 mg/L 63–104% of the *Lactuca* seeds and 95–100% of the *Raphanus* seeds germinated [53]. Analogously, the magnetite NPs were not toxic to duckweed *Lemna minor* growth inhibition assay (EC_50_ > 100 mg/L) as well as in *Daphnia magna* 48 h immobilization assay (EC_50_ > 1000 mg/L) [54].

The EC_50_ value of bare Fe_2_O_3_ NPs to ciliates was not reached at the highest tested concentration (33 mg/L) (Table 3). According to Aruoja et al. [55], ciliates *Tetrahymena thermophila* 24 h viability assay yielded an EC_50_ value of 26 mg/L and the 72 h EC_50_ for algae *Pseudokirchneriella subcapita,* the toxicity of Fe_2_O_3_ NPs in 72 h growth inhibition assay was 1.9 mg/L. In both cases, the toxicity was not due to the solubilisation of NPs (shedding of Fe-ions).

However, the HA-treated MNPs were more inhibitory to *S. alba* (EC_50_ ~ 900 mg/L) that bare MNPs (EC_50_ > 3300 mg/L; Table 3), probably due to the effect of humic acids (EC_50_ ~ 900 mg/L mg/L). The harsh oxidation (but not the mild oxidation) somewhat increased the toxicity of HA-treated MNPs to plants. Despite that, the EC_50_ values of all these tested compounds were > 100 mg/L and could be considered not harmful (Table 3, Figure 3B).

Concerning the effects of MNPs to ciliates, although there was not a very clear dose-effect relationship, the treatment with HAs and following oxidation increased the toxicity of MNPs (Figure 3A). Thus, in general, the trends were similar for plants, but the MNPs, especially after the addition of HAs and following oxidation were remarkably more toxic to ciliates (EC_50_ down to 0.33 mg/L). Thus, the toxicity of iron oxide NPs to ciliates increased in parallel with oxidation of the Fe_3_O_4_ following the order: bare Fe_3_O_4_ < Fe_3_O_4_/HA < Fe_3_O_4_/HA mild oxidation < Fe_3_O_4_/HA harsh oxidation, to *S. alba:* bare Fe_3_O_4_ < Fe_3_O_4_/HA mild oxidation < Fe_3_O_4_/HA < Fe_3_O_4_/HA harsh oxidation. Summing up, the most toxic compound in both assays were HA-treated MNPs after harsh oxidation (Table 3, Figure 3). The latter may be due to the partial destruction of HAs supramolecule units and released ferrous and ferric ions during harsh mechanical treatment of the Fe_3_O_4_/HA as ferrous ions are far more toxic than the inherent toxicity of the iron-containing NPs [34]. The reason for the release of the iron ions from Fe_3_O_4_/HA matrix is probably the supramolecular nature of HA that means associates via weak hydrophobic (van der Waals, π-π, CH-π) and hydrogen bonds [56]. As a result of different treatments (includian oxidation), HA can be hydrolyzed, thus leading to the destruction of the protective layer of Fe_3_O_4_/HA. Derivatives of HA contribute to the iron ions release from magnetite. Both iron ions, Fe^2 +^ and Fe^3+^, form strong mixed ligand complexes with HA [57]. Therefore, thermodynamically driven dissolution and subsequent complexation reactions between HA and iron ions can be important for Fe_3_O_4_/HA dissolution. The dissolution of Fe_3_O_4_/HA was also confirmed by hydrodynamic particle size analysis via DLS showing a decrease in the particle size [58].

The meta-analysis of the ecotoxicity of NPs made by Juganson et al. [35] described Fe-oxide NPs of relatively low toxicity: the toxicity order for seven types of NPs involved in this study was Ag > ZnO > CuO > CeO_2_ > CNTs > TiO_2_ > FeO_x_ (Fe_2_O_3_, Fe_3_O_4_). Notably, FeO_x_ NPs had also the lowest amount of available information on their ecotoxicity compared to other above listed NPs and the information on ecotoxicity of FeOx particles started to emerge in 2009, i.e., later than for the other selected NPs and by the publishing of the paper (2015) there was no data about possible mechanism of action of FeOx NPs.

### 3.4. Structure–Bioactivity Relationship for MNPs

Correlating physicochemical properties of studied NPs (primary size, hydrodynamic size, z-potential, percentage of magnetite) and ecotoxicity of MNPs (Table 3, Figure 4) it can be concluded that humic acids can change the toxicity of MNPs in different ways. Importantly, after each step of treatment, whether it be humic acids, harsh oxidation, or 90 days of aging, the percentage of magnetite in MNPs after treatment with HA decreased from 83% to 79% and further to 48% after the harsh oxidation of Fe_3_O_4_/HA NPs. Interestingly, after mild oxidation of Fe_3_O_4_/HA the percentage of magnetite fell practically to zero (Table 3). The zeta potentials and hydrodynamic diameters of the above described MNPs also changed reflecting the degree of Fe_3_O_4_ NPs surface modification by humic acids and hydrolysis of the humic shell during boxidationsn in harsh and mild conditions (Table 3).

### 3.5. The Ecotoxicity of Fe-Ions before and after Addition of Humic Acids

It is still unclear whether the toxicity of iron oxide NPs is specifically related to nanoparticle properties (such as nano-size that also translates into big relative surface area etc.) or is due to the toxicity of shed Fe^2+^ or Fe^3+^ ions, or, both Juganson et al. [35]. Wang et al. [59] studied the toxicity of Fe_2_O_3_ NPs to bioluminescent bacteria *Photobacterium phosphoreum* and observed EC_50_ value of 200 mg/L (a moderate toxic effect) and showed that the toxic effect was caused solely by NPs and not by shed ions as the NPs were practically nonsoluble. Fe^3+^ ions, however, were very toxic to *P. phosphoreum*, EC_50_ about 0.03 mg Fe/L.

As in the environment the NPs become in contact with humic substances that may affect their behavior and properties, we evaluated the toxicity of Fe_3_O_4_-NPs and iron ions (Fe^2+^ and Fe^3+^) in the presence of humic acids (HA), a natural organic polymer that easily forms complexes with Fe^2+^ and Fe^3+^ and also binds to NPs surface. As test species, plants *S. alba* and ciliates *P. caudatum* were used. The Fe(II, III)-humic complexes and Fe_3_O_4_-HA NPs were synthesized as described in Materials and Methods. Test organisms were exposed to: control—culture medium without Fe and HA; samples—Fe(II) or/and Fe(III), HA, Fe(II, III)HA, Fe(II)HA, Fe(III)HA, Fe(II, III)HA.

#### 3.5.1. Ecotoxicity of Fe^2+^/Fe^3+^

As mentioned above, Fe^3+^ ions were very toxic to bioluminescent bacterium *P. phosphoreum*, EC_50_ about 0.03 mg Fe/L [59]. Analogously, Kurvet et al. [60] showed high toxicity of Fe^3+^ ions to another luminescent bacterium, *Vibrio fischeri*, EC_50_ = 0.44 mg/L and EC_50_ of 4.9 mg/L was obtained for ciliates *T. thermophila*. Somewhat lower toxicity of Fe^3+^ ions was shown towards algae *P. subcapitata* (EC_50_ = 23 mg/L) by Aruoja et al. [55].

The current study shows that the EC_50_ values of the Fe-ions to ciliates ranged from 1–3 mg/L whereas Fe^3+^ ions were slightly less toxic than Fe^2+^ ions (Figure 5A) and to *S. alba* from 35–60 mg/L and again, Fe^3+^ ions were slightly less toxic than Fe^2+^ ions (Figure 5B, Table 4). Lower Fe^3+^ toxicity to plants may be due to their lower bioavailability [61] as Fe^2+^ is more soluble and can be more easily absorbed by plants [62]. In general, Fe^2+^ and Fe^3+^ ions were about tenfold more toxic to ciliates than to plants *S. alba*. The mixture of Fe^2+^/Fe^3+^ showed different type of dose-response compared to Fe^2+^ and Fe^3+^ ions separately when exposed to ciliates (Figure 5A), but not in case of *S. alba*. The difference in dose-response may be a result of contribution of several species of ferrous and ferric ions (aqua-ions Fe^2+^, Fe^3+^ and hydroxocomplexes FeOH^+^, Fe(OH)^2+^) due to their hydrolysis in biotesting environment used for ciliates.

#### 3.5.2. Effects of Humic Acids on the Ecotoxicity of Fe^2+^ and Fe^3+^

The addition of humic acids to iron ions reduced their toxicity to *S. alba* about 10-fold and EC_50_ values of Fe-HA complexes proved higher than 100 mg/L, i.e., the complexes could be considered not harmful. The same trend was observed in case of toxicity to ciliates: the addition of HA decreased the toxicity of Fe-ions about 10-fold. However, Fe^3+^/HA still remained toxic (EC_50_ about 25 mg/L) (Figure 5, Table 4). The HA showed also some inhibitory effect to both, ciliates and plants but in general their EC_50_ values to plants were not exceeding 100 mg/L (Figure 5B) and most probably the same would have been true for ciliates although so high concentration was not tested (Figure 5A).

The addition of humic acids to binary iron solutions lead to a significant reduction in toxicity (EC_50_ changed from 0.5 to > 100 mg Fe/L) probably due to the formation of chelate-type ternary iron ions humic acids complexes.

Mitigating effect of HA in Fe-ions and HA complexes could be explained by the fact that Fe(II) and Fe(III) are strongly bound to acidic functional groups of HAs [63,64,65,66,67]. It has been reported that complexes of Fe^2+^ and Fe^3+^ with humic acids increased Fe-uptake in plants Garcia [38]. These results are coherent with the fact that after the addition of humic acids Fe^2+^ and Fe^3+^ became less toxic to plants (Table 4). Thermodynamic constants for 1:1 complexes of Fe(III) with weak binding sites in humic substances like carboxylic groups and strong binding sites like phenolic groups are four to 40 times greater than the respective stability constants for Fe(II) complexes [68]. That could be a reason for more pronounced mitigating effect of HA in case of Fe^3+^ (Table 4).

### 3.6. Acute Toxicity of Fe^2+^/Fe^3+^ with and without Addition of Humic Acids: Effect of Aging

The influence of the aging on the toxicity of HA, Fe^2+^/Fe^3+^ and Fe(II,III)HA was studied using *S. alba* and *P. caudatum* as test species (Table 5, Figure 6). To simulate the aging in mild conditions, the preparations were incubated in water in the dark at 5 °C during 90 days.

In case of ciliates *P. caudatum* (Figure 6A), there were no statistically significant differences in toxicity of as-prepared and aged Fe^2+^/Fe^3+^: both preparations were ‘very toxic’ (EC_50_ < 1 mg/L). However, the aging had pronounced effect on toxicity to ciliates in case of humic acids as well as Fe-ions modified with HAs. In both cases the toxicity remarkably increased after aging (Table 5). Interestingly, the toxicity of aged Fe(II,III)HA (EC_50_ = 1.4 mg Fe/L) was higher than the toxicity of HA (EC_50_ = 1.93 mg/L). The increase of toxicity can be explained by the formulation of hydroxocomplexes of iron ions with humic acids, as well as hydrolysis of humic acids with decomposition into low molecular fractions and fulvic acids, that may affect *P. caudatum.*

For the *S. alba* (Figure 6B), the Fe^2+^/Fe^3+^ proved not toxic before and after aging for 90 days (EC_50_ 75 and 231 mg/L, respectively) and the addition of HAs further decreased their inhibitory effect to plants (EC_50_ of Fe(II,III)HA > 900 mg/L) with no remarkable effect of aging (EC_50_ = 915 mg/L)(Table 5).

## 4. Conclusions

In the current study, the magnetite NPs (MNPs) were studied for ecotoxicity (ciliates and plants were used as test species) before and after addition of humic acids, to obtain new scientific information on toxicity of magnetite NPs in environmentally relevant conditions. As Fe-ions maybe shed from MNPs and these ions are far more toxic than MNPs, their toxic effects were studied in parallel. In general, MNPs and Fe-ions were remarkably more inhibitory to ciliates *Paramecium caudatum* than to plants *Sinapis alba*. The MNPs (with or without HA) had no remarkable inhibitory effects to plants *S. alba* even after harsh oxidation (96 h EC_50_ ~ 300 mg/L). However, Fe-ions were toxic to *S. alba* (EC_50_ 25–60 mg/L) but the toxic effect was mitigated after addition of HAs.

To ciliates MNPs were toxic whereas the complexing with HAs and aging increased the toxic effects down to 24 h EC_50_ = 0.33 mg/L after harsh oxidation. The toxicity of Fe-ions to ciliates (EC_50_ 1–3 mg/L) was reduced after addition of HAs about 10-fold but the complexes still remained toxic to ciliates. Thus, the addition of HAs and aging increased the toxicity of MNPs and mitigated the toxic effect of Fe-ions.

The changes in toxicity were accompanied with the changes in physicochemical properties of MNPs: after each step of treatment as by humic acids, harsh oxidation or 90-days aging the percentage of stoichiometric magnetite decreased from 83 to 79%, to 48%, and to practically to zero, respectively. The zeta potential and hydrodynamic diameter of the above described MNPs also changed reflecting the degree of Fe_3_O_4_ NPs surface modification by humic acids and hydrolysis of humic shell during both oxidation in harsh and mild conditions.

## Figures and Tables

**Figure 1 nanomaterials-10-02011-f001:**
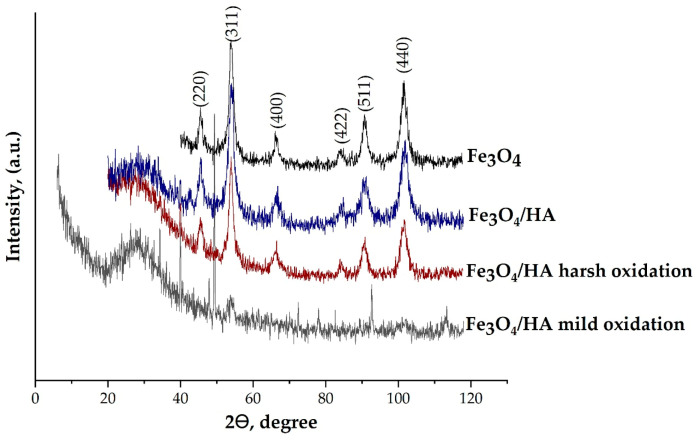
XRD patterns of synthesized Fe_3_O_4_ nanoparticles. HA—humic acids.

**Figure 2 nanomaterials-10-02011-f002:**
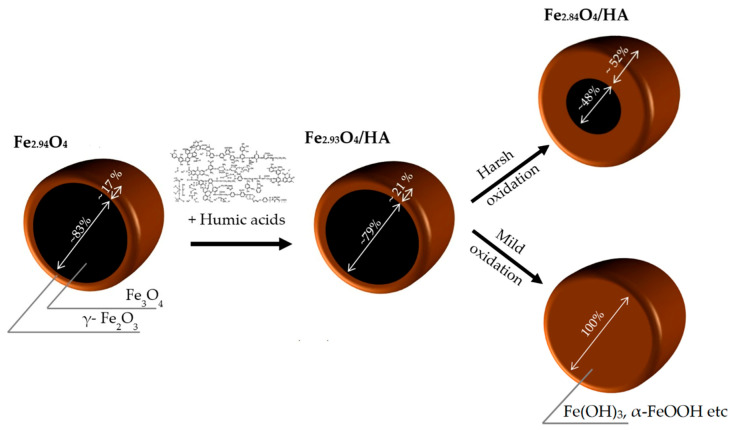
Schematic view of the oxidative transformation of Fe_3_O_4_ (magnetite; black) to maghemite (Fe_2_O_3_; brown) by treatment with humic acids followed by mild or harsh oxidation. See also Table 1.

**Figure 3 nanomaterials-10-02011-f003:**
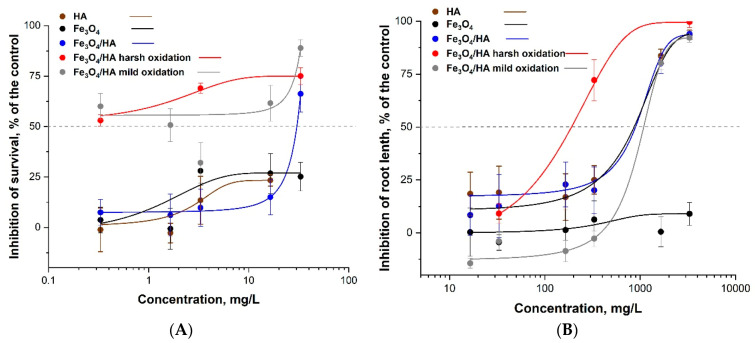
Dose–effect relationships of aqueous suspensions of Fe_3_O_4_ NPs (MNPs) and MNP-HA complexes before and after harsh and mild oxidation in (**A**) ciliates *Paramecium caudatum* 24 h immobilization test and (**B**) higher plants *Sinapis alba* 96 h root growth inhibition test: A dose-effect study. All concentrations are nominal. HA–humic acids. Average values ± SD of triplicates.

**Figure 4 nanomaterials-10-02011-f004:**
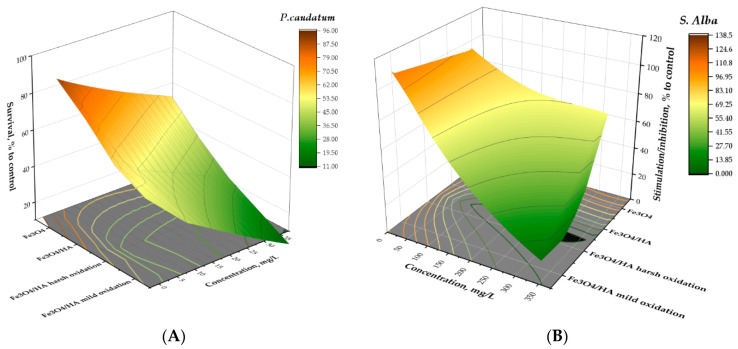
3D-surface for dependence of toxicity on modification of Fe_3_O_4_ MNPs; (**A**) ciliates *Paramecium caudatum* 24 h immobilization test and (**B**) higher plants *Sinapis alba* 96 h root growth inhibition test: A dose-effect study. All concentrations are nominal.

**Figure 5 nanomaterials-10-02011-f005:**
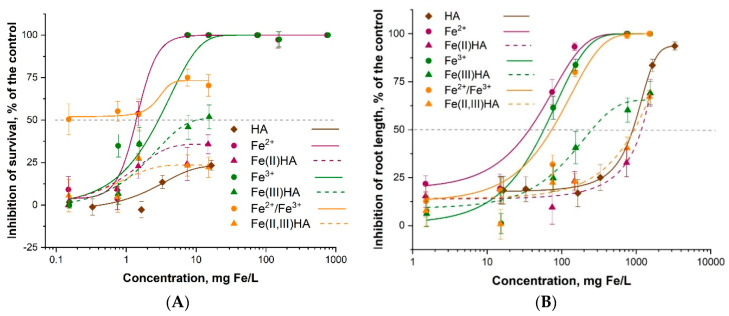
Dose–effect relationships of aqueous suspensions of humic acids (HA) and iron ions with. and without the added humic acids (**A**) in ciliates *Paramecium caudatum* 24 h immobilization test and (**B**) in plants *Sinapis alba* 96 h root growth inhibition test. Average values ± SD of triplicates. All concentrations are nominal.

**Figure 6 nanomaterials-10-02011-f006:**
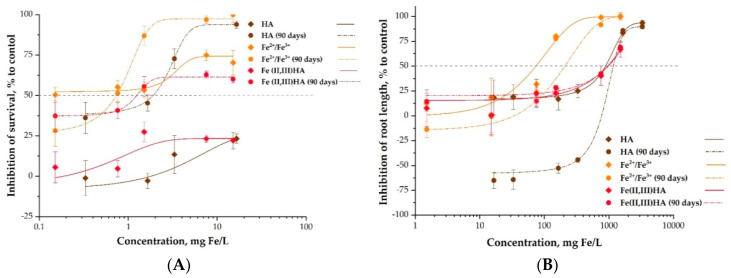
Dose–effect relationships of aqueous suspensions before and after storage (**A**) in ciliates *Paramaecium caudatum* 24 h immobilization test and (**B**) in plants *Sinapis alba* 96 h root growth inhibition test. Average values ± SD of triplicates. All concentrations are nominal and concentrations of HA are calculated as mg of HA/L.

**Table 1 nanomaterials-10-02011-t001:** Microstructural parameters of synthesized magnetite nanoparticles (MNPs). HA—humic acids.

Sample	Fe_3_O_4_		Fe_3_O_4_/HA		Fe_3_O_4_/HA Harsh Oxidation		Fe_3_O_4_/HA Mild Oxidation	
hkl	2θ, °	d, Å	2θ, °	d, Å	2θ, °	d, Å	2θ, °	d, Å
220	45.60	2.971	45.67	2.966	45.62	2.954861	49.33	2.74502038
311	53.98	2.535	54.01	2.535	53.92	2.526709	53.92	2.52670925
400	66.28	2.094	66.36	2.094	66.39	2.09233	-	-
422	84.25	1.714	84.25	1.714	84.38	1.705695	82.6	1.73564713
511	90.77	1.610	90.71	1.612	90.62	1.61133	92.69	1.58329633
440	101.52	1.476	101.4	1.476	101.68	1.477368	101.3	1.48137719
a, A	8.383(2)		8.382(6)		8.365(5)		8.250(8)	
x	0.387(7)		0.382(1)		0.232(7)		-	
δ	0.059(4)		0.062(7)		0.154(2)		-	
Fe_3-δ_O_4_	Fe_2.94_O_4_		Fe_2.93_O_4_		Fe_2.84_O_4_		-	
% Fe_3_O_4_	~82.7		~79.2		~48.3		0	
D, nm	6.9 ± 2.4		10.3 ± 1.3		7.8 ± 1.9		11.03 ± 5.1	
CV, %	34		12.6		24.6		50.2	

% Fe_3_O_4_ is the content of stoichiometric magnetite, CV—coefficient of variation.

**Table 2 nanomaterials-10-02011-t002:** Magnetic properties of bare, modified and oxidized magnetite NPs.

Sample	Saturation Magnetization Ms, emu/g	Remanent Magnetization Mr, emu/g	Coercive Force Hc, Oe
Fe_3_O_4_ (bare)	68.2	6.88	74.1
Fe_3_O_4_/HA	30.9	6.40	160
Fe_3_O_4_/HAox (harsh oxidation)	15.7	4.11	159

HA—humic acids, ox—oxidized.

**Table 3 nanomaterials-10-02011-t003:** Physicochemical parameters of MNPs and ecotoxicity (EC_50_ *, mg/L) of MNPs and humic acids (HA).

Sample	%Fe_3_O_4_	ξ, mV	Hydrodynamic Diameter, nm	24 h EC_50_ for *P. caudatum*, mg/L	96 h EC_50_ for *S. alba,* mg/L
Fe_3_O_4_	83	−25.5 ± 6.03	400.8	>33 **	>3300
HA	-	-	-	>33 **	900.65 ± 106.2
Fe_3_O_4_/HA	79	−38.8 ± 7.1	153.2	32 ± 3.1	902.71 ± 211.5
Fe_3_O_4_/HA harsh oxidation	48	−16.19 ± 2.1	886.7	0.33 ± 0.01	330 ± 10.1
Fe_3_O_4_/HA mild oxidation	0 *	−49.63 ± 2.2	586.1	1.40 ± 0.5	951.6 ± 80.2

* EC_50_ is the concentration of a sample reducing the root length or survival of ciliates by 50%; ** highest concentration that was tested; Color code: ≤1 mg/L (red  )= very toxic; >1-10 mg/L (orange  ) = toxic; >10–100 mg/L (yellow  ) = harmful; >100 mg/L (green  ) = “not classified/not harmful”. EC_50_ data not allowing ranking are on white background.

**Table 4 nanomaterials-10-02011-t004:** The toxicity (EC_50_) of magnetite nanoparticles (MNP)s, humic acids (HA), Fe^2+^ and Fe^3+^ ions and their complexes with HAs to ciliates *P. caudatum* and plants *S. alba.*

Sample	EC_50_ ± SD (mg/L)
Toxicity to ciliates *P. caudatum* (24 h EC_50_)	
Fe^2+ a^	1.15 ± 1.4
Fe^3+ a^	3.27 ± 0.8
Fe^2+^/Fe^3+ c^	0.48
Fe_3_O_4_	>33
HA	>33
Fe(II)HA	>33
Fe(III)HA	24.6 ± 9.3
Fe(II,III)HA ^b,c^	>15.1
Fe_3_O_4_/HA ^b^	32 ± 3.1
Toxicity to plants *S. alba* (96 h EC_50_)	
Fe^2+^	35.92 ± 18.6
Fe^3+^	58.71 ± 21.2
Fe^2+^/Fe^3+^	75.21 ± 44.4
Fe_3_O_4_ ^c^	>3300
HA	900.65 ± 106.2
Fe(II)HA	1402.3 ± 135.1
Fe(III)HA	256.18 ± 50.4
Fe(II,III)HA	910.43 ± 270.1
Fe_3_O_4_/HA	902.71 ± 211.5

All concentrations are nominal (mg of compound L^−1^). Average values ± SD of triplicate; ^a^ Values calculated with IBM SPSS Statistica using the probit method; ^b^ Values calculated with R using ANOVA. a-c indicate statistical significance. Values with the same letters (a or b) are not significantly different (*p* > 0.05). Samples with letter “c” have no significantly different at concentrations. EC_50_ values are calculated as mg/L for Fe_3_O_4_, HAs and Fe_3_O_4_/HA and as mg Fe/L for Fe-ions; Color code: ≤1 mg/L (red  )= very toxic; >1–10 mg/L (orange  ) = toxic; >10–100 mg/L (yellow  ) = harmful; >100 mg/L (green  ) = “not classified/not harmful”. EC_50_ data not allowing ranking are on white background.

**Table 5 nanomaterials-10-02011-t005:** The toxicity (EC_50_
^a^) of humic acids (HA), as prepared and aged Fe^2+^/Fe^3+^ ions to *P. caudatum* and *S. alba.*

Sample	EC_50_ ± SD(mg Fe/L)
**Toxicity to ciliates *P. caudatum* (24 h EC_50_)**
HA *	>33
HA * (90 days)	1.93 ± 0.55
Fe^2+^/Fe^3+^	0.48 ± 0.02
Fe^2+^/Fe^3+^ (90 days)	0.67 ± 0.02
Fe(II,III)HA	>15.1
Fe(II,III)HA (90 days)	1.4 ± 0.4
**Toxicity to plants *S. alba* (96 h EC_50_)**
HA *	900.65 ± 106.2
HA * (90 days)	880.1 ± 123.1
Fe^2+^/Fe^3+^	75.21 ± 44.4
Fe^2+^/Fe^3+^ (90 days)	231.41 ± 97.46
Fe(II,III)HA	910.43 ± 270.1
Fe(II,III)HA (90 days)	915.23 ± 145.2

All concentrations are nominal (mg of weighed compound L^−1^). Average values ± SD of triplicate; ^a^ Values calculated with IBM SPSS Statistica using the probit method; ^b^ Values calculated with R using ANOVA; * EC_50_ values for HA are calculated as mg/L; Color code: ≤ 1 mg/L (red  ) = very toxic; > 1–10 mg/L (  ) = toxic; >10–100 mg/L (yellow  ) = harmful; > 100 mg/L (green  ) = “not classified/not harmful”. EC_50_ data not allowing ranking are on white background.

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
