# Peer review of "Effects of Humic Acids on the Ecotoxicity of Fe3O4 Nanoparticles and Fe-Ions: Impact of Oxidation and Aging"

_nanomaterials, 2020, doi:10.3390/nano10102011_

Round 1

Reviewer 1 Report

The manuscript is focused on the changes of the ecotoxicity of MNPs in dependence on different factors. This interdisciplinary study describes the mechanisms of the changes occurred in MNPs and Fe(II)/Fe(III), with humic acids and without them, after oxidation and aging. The results demonstrated the [combination/treatment]-specific changes of ecotoxicity. This study contributes to addressing a knowledge gap in ecotoxicology of nanomaterials under real variable conditions. The manuscript clearly describes objectives. The research is important and novel to the field. The Introduction is clear and well organized. The description of methods is accurate. The Results section represent well-structured data. Colored tables bring a better visualization of results. Conclusions summarize the importance of the data obtained in this study.

Some minor remarks are listed below:

L22-23. Abstract. Please specify the period of testing, e.g., EC50 (24h).

L235. Equation (1). It is advisable to place this equation in Methods.

L308-314. This block would be more suitable to place after description of own results.

L336. “Gradulal oxidation” is not appropriate here. Two types of treatments completely different results (Fig.2).

L352. Caption. Harsh and mild oxidation.  

L387. “:” – TYPO?

Fig.5A. X axis are extended more than 33 mg/L. Please correct.

Kind regards

Author Response

Dear Reviewer, 

Thank you for your detailed review of our paper. Please find enclosed the revision’ details: line numbers, notes and Reply/Correction.

Reviewer 2 Report

General comments

The ms shows interesting results about the effect of humic acids on the ecotoxicity of of Fe3O4 NPs and Fe ions. Aspects related to the effect of oxidation and aging in the NPs ecotoxicity are considered. The main weakness of this ms is related to the fact that ecotoxicity aspects are focused only in two species target, Paramecium caudatum (ciliate) and Sinapis alba (white mustard). Another species -more representative of aquatic environment- will be convenient to include. Information about environmental risk and ecological relevance of data should be included.  This is well carried out study where the properties of NPs and the efect of HA are analized, but ecological relevance is not assessed.

Specific comments

Abstract summarizes well the main results.

Introduction

It is well written and structured

M&M

Can you give additional information the reason of the selected conditiond for  mild and harsh.

What is the reason to show the results as nominal concentration? (section 2.5, line 169).

Results and Discussion

As it was mentioned previously, information about the occurrence of these NPs in the environment can be helpful and it would be useful to know the relevance of the risk associated to use of these nanomaterials (bare and aging).

Author Response

(The authors gave the same response as above.)
